# An Anatomical-Based Subject-Specific Model of In-Vivo Knee Joint 3D Kinematics From Medical Imaging

**Fabrizio Nardini** [1], **Claudio Belvedere** [2,*], **Nicola Sancisi** [1], **Michele Conconi** [1], **Alberto Leardini** [2], **Stefano Durante** [3] **and Vincenzo Parenti-Castelli** [1]

[1] Department of Industrial Engineering, University of Bologna, Viale Risorgimento 2, 40136 Bologna, Italy; fabrizio.nardini2@unibo.it (F.N.); nicola.sancisi@unibo.it (N.S.); michele.conconi@unibo.it (M.C.); vincenzo.parenti@unibo.it (V.P.-C.)

[2] Movement Analysis Laboratory, IRCCS Istituto Ortopedico Rizzoli, Via di Barbiano 1/10, 40136 Bologna, Italy; leardini@ior.it

[3] Nursing, Technical and Rehabilitation Assistance Service, IRCCS Istituto Ortopedico Rizzoli, Via di Barbiano 1/10, 40136 Bologna, Italy; stefano.durante@ior.it

\* Correspondence: claudio.belvedere@ior.it; Tel.: +39-0516366570, Fax: +39-0516366561

**Abstract:** Biomechanical models of the knee joint allow the development of accurate procedures as well as novel devices to restore the joint natural motion. They are also used within musculoskeletal models to perform clinical gait analysis on patients. Among relevant knee models in the literature, the anatomy-based spatial parallel mechanisms represent the joint motion using rigid links for the ligaments' isometric fibres and point contacts for the articular surfaces. To customize analyses, therapies and devices, there is the need to define subject-specific models, but relevant procedures and their accuracy are still questioned. A procedure is here proposed and validated to define a customized knee model based on a spatial parallel mechanism. Computed tomography, magnetic resonance and 3D-video-fluoroscopy were performed on a healthy volunteer to define the personalized model geometry. The model was then validated by comparing the measured and the replicated joint motion. The model showed mean absolute difference and standard deviations in translations and rotations, respectively of $0.98 \pm 0.40$ mm and $0.68 \pm 0.29°$ for the tibia–femur motion, and of $0.77 \pm 0.15$ mm and $2.09 \pm 0.69°$ for the patella–femur motion. These results show that accurate personalized spatial models of knee kinematics can be obtained from in-vivo imaging.

**Keywords:** knee joint; kinematic model; subject-specific; in vivo; medical imaging data

## 1. Introduction

Kinematic models of the knee aim at the description of the natural motion of the joint and have many clinically relevant applications, such as articular surface and ligament reconstructions, pre-operative plans [1], gait analysis and soft–tissue artifact compensation through multibody optimization techniques [2]. Many kinematic models of the knee were proposed [2,3], most of them focusing on the tibio–femoral (TF) joint. Some of these models represent the knee joint as a rigid-link one-degree-of-freedom (1-DoF) mechanism, based on the experimental evidence that, in unloaded conditions, the joint motion exhibits 1-DoF and the joint passive structures show a minimal change in shape or length. Starting from planar models like the hinge and the four–bar linkage [4,5] these representations evolved into spatial models [6–8] guided by ligaments and articular surfaces [4,8–12]. These spatial models were able also to describe the out-of-plane components of the knee motion, by representing the connection between anatomical structures and joint motion in 3D. Based on

the same principles, three-dimensional kinematic models of the knee joint including also the patello–femoral (PF) joint were proposed [13,14].

All these models were mainly defined and validated in vitro, using techniques hardly exploitable in vivo in the standard clinical practice, like dissection for anatomical geometry and bone pins for kinematic measurements. However, great attention was devoted recently to develop subject-specific medical procedures and devices customized on the single patient [1,15,16]. Therefore, the definition of kinematic models that reproduce the anatomy and the motion of the specific patient in vivo is gathering a growing importance. In this case, personalization and identification of the model geometry are the main problems, since only non-invasive measures can be performed in vivo, and these often present limitations on the quality, type and quantity of the acquired data.

A common practice is scaling a generic model to fit specific dimensions of the single subject using corresponding anatomical landmarks [17–20]. Nevertheless, scaling considers only the gross anthropometry of the subject, disregarding all its distinctive features (ligament attachment positions on the bones, articular surface shapes). Eventually, the final model does not reproduce either the anatomy or the kinematics of the joint and thus it is not actually personalized [21].

An alternative approach is using medical images to personalize the model geometry on the subject anatomy. However, model customization in vivo is not straightforward, particularly for complex spatial models due to the higher number of parameters to be identified and the increased sensitivity to their variations [22,23]. For these reasons, as a recent review demonstrates [2], simplified representations of the knee joint such as the hinge and the four-bar linkage are often preferred. Whereas these models have the advantage of being based on fewer parameters, they are far from an accurate 3D representation of the behaviour of the knee joint and of its anatomical structures [20,24].

Several studies investigated the importance of anatomical accuracy of joint models in musculoskeletal modelling [25,26], concluding that the use of models with a strong anatomical foundation leads to more reliable estimations of motion and forces (i.e., joint reactions, ligament and musculo–tendinous forces) during gait. Nevertheless, only few subject-specific kinematic knee models with such features exist for in vivo applications. A first approach prescribes an experimentally measured joint motion within the musculoskeletal model, like, for instance, in the seminal work of Yamaguchi and Zajac [27]. This approach has limitations: due also to experimental measurement inaccuracies, the motion is not consistent with joint constraints and unphysiological ligament elongations and cartilage compenetration can be observed; moreover, there is not an explicit representation of the joint constraints, so their behaviour under different loading conditions cannot be determined. A second approach tries to add an explicit representation of the joint constraints. In a 1-DoF spatial knee joint model based on a specific subject [28], only two out of five motion parameters depended on the actual knee geometry, while the other three were related to the flexion angle using average generic functions. Parallel mechanisms showed the connection between all motion parameters of both the TF and PF joint to the actual knee geometry through a one-to-one replica of the primary joint constraints [7,11,14]. However, they were mainly defined in vitro. A recent study defined a spatial parallel mechanism in vivo [16], though the proposed procedure was based on average kinematic data from the literature, and no corresponding validation work was performed.

To overcome the limitations of the aforementioned studies, the aim of this investigation is to define and validate a complete and accurate procedure to obtain a subject-specific 3D kinematic model of the knee in vivo, by combining the two approaches described above. The idea is using an experimental motion measured on the subject, adding at the same time an explicit representation of the joint constraints in order to overcome the limitations of a direct application of the experimental measurements in the musculoskeletal model. The procedure investigates the use of standard medical imaging techniques to tailor the geometry of the model including both the TF and PF joints to the subject anatomy. Validation is assessed by comparing the observed and predicted joint motions of a volunteer.

## 2. Methods

### 2.1. Data Acquisition

A healthy subject (male, 66 years old, 169 cm, 58 kg) with no history of knee joint diseases volunteered for the study. The present study was approved by the ethical committee at the Istituto Ortopedico Rizzoli, and the volunteer gave his informed consent.

Computed tomography (CT) (Brilliance CT 16-slice system, Philips Healthcare, Best, Netherlands) and magnetic resonance imaging (MRI) (1.5 Tesla MRI; SIGNA EXCITE HDxt, GE Healthcare, Chicago, IL, USA) scans were performed. With the subject supine, local CT scans of the right knee (220 × 220 mm, 512 × 512 pixels, 2 mm slice thickness) were taken in the transverse plane. MRI scans of the same knee were collected in both the sagittal, coronal and axial planes (220 × 220 mm, 512 × 512 pixels, 0.8 mm slice thickness) using Sagittal Cube PD, Coronal Cube PD, Axial PD Fast Spin Eco sequences, respectively. To define the anatomical reference systems for the bone models, local CT scans of the right ankle and hip joints (220 × 220 mm, 512 × 512 pixels, 5 mm slice thickness) in the transverse plane and a CT scout view of the whole lower limb at full extension (1120 × 500 mm, 1128 × 512 pixels) in the frontal plane were taken as well.

The experimental natural knee motion was reconstructed using single plane 3D video-fluoroscopy (3DV) by means of a digital remote-controlled diagnostic radiological device with flat-panel detector (CAT Medical Systems, Monterotondo, Italy). According to an established protocol [29], the reconstruction procedure starts with the collection of a series of digital video-fluoroscopic images of the knee joint in the sagittal plane: frequency was set at 15 frame/s and the exposure parameters were set initially at 50 mA and 60 kV and then adjusted automatically by device-based automatic exposure control. The 430 × 430 field of view (FOV) was set to record the proximal half of the tibia and the distal half of the femur, whereas the 3D geometry of the fluoroscopy device (principal distance and principal point), as well as the definition of the reference frame of the detection plane, were derived from images of calibration objects [30], these being manufactured via numerical control with known specific geometrical features. The volunteer was asked to perform a knee flexion–extension movement starting at full extension. The task was performed with the volunteer supine, the right lower limb lying on a smooth plane, by pulling the leg towards the chest up to full flexion with the aid of a lace tied around the thigh, and then by letting the leg extend back to full extension. The whole movement lasted about 4 s. A wedge-shaped support was placed behind the subjectý back to prevent thigh muscles contraction during the test. The heel of the subject was placed on a polytetrafluoroethylene disk allowed to slide smoothly on the plane to reduce friction. Two acquisitions of a complete knee motion cycle were performed, since the FOV of the fluoroscope did not allow the motion to be recorded in one take. The first acquisition recorded the range full extension, 80° and back to full extension, while the second recorded the range from 80° to full flexion and back to 80°.

### 2.2. Image Data Post-Processing

Manual segmentation was performed on both the CT and MRI images. 3D bone surfaces of the proximal and distal femur, proximal tibia, patella and malleoli were obtained from the CT. A second 3D representation of anatomical surfaces also featuring cartilage of the femur and tibia, and the attachment areas of each ligament involved in the kinematic model was obtained from MRI. Performances and accuracy assessment of the whole procedure of morphological reconstruction using the adopted imaging acquisition devices were reported in a recent study [31].

Anatomical reference systems of the femur ($S_f$), tibia ($S_t$) and patella ($S_p$) were defined on the CT-based bone surfaces, since the bony anatomical landmarks could be identified more accurately, through virtual palpation (Figure 1a). The MRI-based surfaces, including bone and cartilage, were aligned to the CT–based ones by selecting corresponding bony landmarks (Figure 1b), used as fiducials, and by aligning them using a least-square algorithm. Fiducials were chosen among the

most distinguishable bony features both on the MRI and CT images. The result of the alignment was visually inspected to exclude possible errors in the landmark identification.

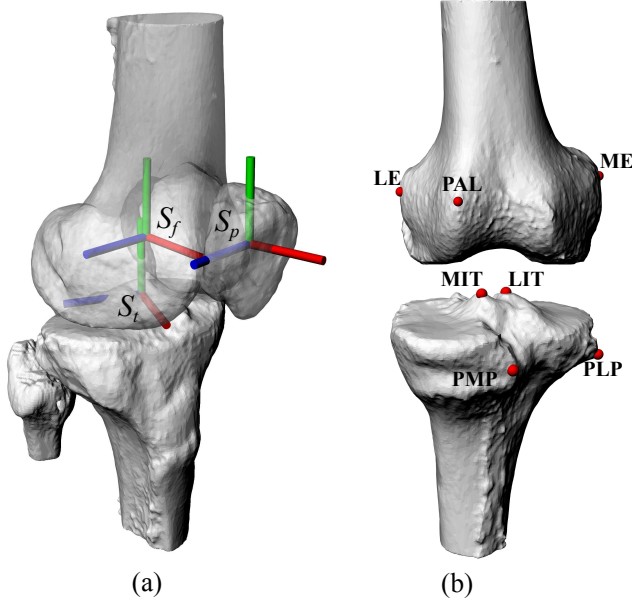

**Figure 1.** Figure (**a**) shows an antero–lateral view of a complete knee with the anatomical reference systems $S_t$, $S_f$ and $S_p$ [14] superimposed: the red, green and blue axes are the x, y and z axes respectively. Figure (**b**) shows the landmarks utilized for the alignment of the bone models obtained from computed tomography (CT) and magnetic resonance imaging (MRI). To ease the identification of the landmarks, a posterior view of the tibia and an anterior view of the femur are shown. On the femur: lateral epicondyle (LE), medial epicondyle (ME), peak of the lateral anterior ridge (PAL). On the tibia: lateral intercondylar tubercle (LIT), medial intercondylar tubercle (MIT), most posterior point on posterior aspect of the medial compartment (PMP), most lateral point on the posterior aspect of the lateral compartment (PLP).

For 3DV analysis, the CT–based bone surfaces of the femur, tibia and patella were manually registered on the collected video-fluoroscopy images using a dedicated software [30]. The relative three-dimensional bone poses were iteratively checked and corrected to reduce bone–related contour malpositioning and interpenetrations, thus obtaining a reference motion compliant with the volunteer´ŷ articular surfaces. Previous validation works showed an accuracy in knee kinematic reconstruction within 0.5 mm and 1 deg in the sagittal plane [30,32]. In this way, the experimental natural motion of the TF (i.e., the tibia–femur motion) and PF (i.e., the patella–femur motion) joints were obtained [33]. Registration was performed for both the flexion and extension movements, obtaining two motion curves: the volunteer unloaded motion was determined as the mean between the two. Joint rotations at both TF and PF was parametrized according to a standard convention [34] and expressed through the rotation angles $\alpha$ (TF or PF joint flexion[+]/extension[−]), $\beta$ (adduction[+]/abduction[−]) and $\gamma$ (internal[+]/external[−] rotation). The position of the origin of $S_t$ in $S_f$ and of $S_p$ in $S_f$ represented the anterior[+]/posterior[−], proximal[+]/distal[−] and lateral[+]/medial[−] displacements of the TF and PF joints respectively.

### 2.3. Knee Kinematic Model

The kinematic model of the knee [13,14] is a 1-DoF spatial mechanism (Figure 2) composed of two groups of bodies and constraints (i.e., the TF and PF groups), representing the TF and the PF joint respectively. The TF group features three rigid links representing the isometric fibres (IFs) of the anterior cruciate (ACL), posterior cruciate (PCL) and medial collateral (MCL) ligaments, as defined below, and two sphere–sphere pairs representing the medial (MC) and lateral (LC) articular contacts between

the femoral condyles and the tibial plateau. Each sphere–sphere contact can also be represented through a rigid link connecting the sphere centres, since the imposed constraint is equivalent (Figure 2a). Thus, five rigid links model the TF (Figure 2c) connecting five points on the tibia and five points on the femur. The PF group (Figure 2b–d) features a helical pair representing the articular contact between the patella and trochlea, and a rigid link between the tibia and patella representing the patellar tendon (PT).

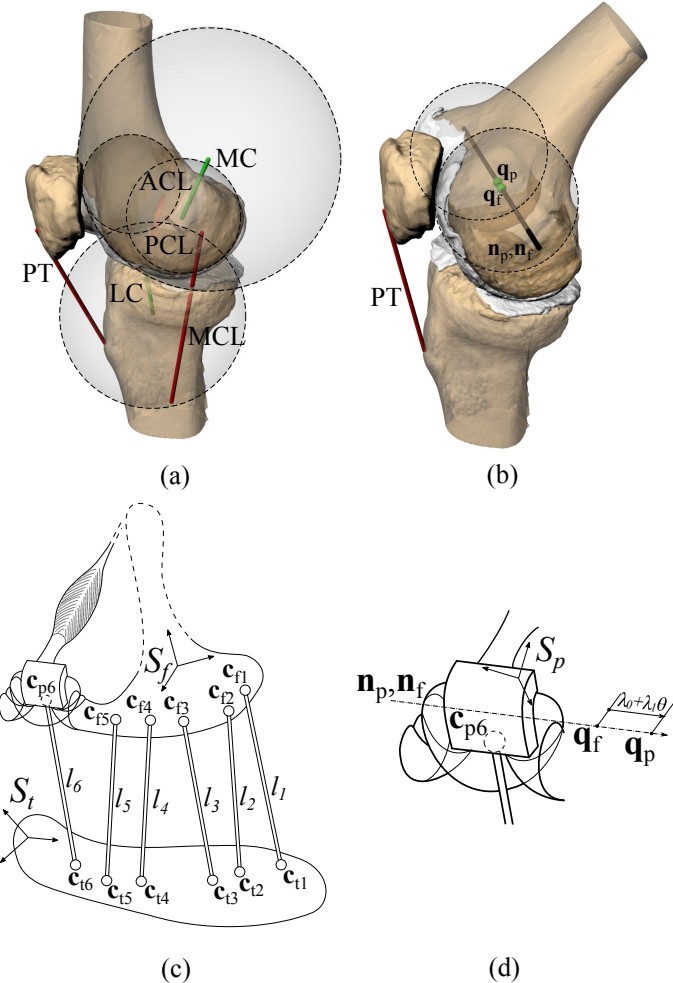

**Figure 2.** The kinematic model of the knee joint. Figures (**a**,**b**) show the anatomical derivation of the tibio–femoral group (TF group) and the patello–femoral group (PF group); Figures (**c**,**d**) show the analytical scheme of the whole knee model and a close up of the PF group, together with the model geometrical parameters.

### 2.4. Definition and Validation of the Subject-Specific Model

The model geometry is defined by 35 parameters for the TF group and 17 for the PF group (Table 1), measured on the CT- and MRI-based surfaces. All parameters were described in the corresponding reference frame. As for the TF, the ends of the two rigid links representing LC and MC (green bars in Figure 2a; $c_{ti}$–$c_{fi}$, $i = 1, 2$, in Figure 2c) were the centres of the spheres obtained from the MRI-based surfaces by a least squares fitting of the posterior part of the femur condylar surfaces and of the medial and lateral facets of the tibial plateau. The three rigid links representing the ACL, PCL, MCL (red bars in Figure 2a; $c_{ti}$–$c_{fi}$, $i = 3, 4, 5$, in Figure 2c) were defined by determining the IF of each ligament. The attachment areas were obtained by projecting the MRI-based ligament attachments on the CT-based bone surfaces. The IF of each ligament was then obtained by computing the couple of points in these areas whose distance showed the smallest variation during the experimental knee motion. To avoid anatomical inconsistencies also accounting for uncertainties in the acquired motion, the search for the

IFs was limited to the anteromedial, the posteromedial and the anterior bundles respectively of the ACL, PCL, and MCL, that showed to be the most isometric ones [9]. The link lengths $l_i$, $i = 1, \ldots, 5$, were the distances between $\mathbf{c}_{ti}$ and $\mathbf{c}_{fi}$ at full extension.

**Table 1.** The geometry of the subject-specific model as directly identified from the medical imaging data (i.e., the first estimation) and after optimization (i.e., the optimized model). Points are represented in the reference system of the corresponding bone. $\mathbf{q}_f$ and $\mathbf{q}_p$ are the intersection points between the patella axis and the sagittal plane, thus $z = 0$ for both vectors, $\mathbf{n}_f$ and $\mathbf{n}_p$ are parametrized by two angles $\theta_k$ and $\phi_k$ $(k = f, p)$: $\mathbf{n}_k = [\cos(\phi_k)\sin(\theta_k)\ \ \sin(\phi_k)\sin(\theta_k)\ \ \cos(\theta_k)]^T$ ([14]).

| Tibio–Femoral Group | First Estimation | | | Optimized Model | | |
|---|---|---|---|---|---|---|
| **ACL Tibia insertion** (mm) $[x_t\,y_t\,z_t]$ | 12.38 | −2.30 | −4.01 | 14.03 | −1.21 | −3.72 |
| **ACL Femur insertion** (mm) $\left[x_f\,y_f\,z_f\right]$ | −8.11 | −3.37 | 7.21 | −8.60 | −1.43 | 7.28 |
| **PCL Tibia insertion** (mm) $[x_t\,y_t\,z_t]$ | −15.59 | −12.43 | 16.68 | −17.24 | −12.15 | 15.60 |
| **PCL Femur insertion** (mm) $\left[x_f\,y_f\,z_f\right]$ | −3.86 | −9.79 | −6.64 | −3.66 | −7.93 | −5.94 |
| **MCL Tibia insertion** (mm) $[x_t\,y_t\,z_t]$ | 5.54 | −73.62 | −17.61 | 4.55 | −72.62 | −16.23 |
| **MCL Femur insertion** (mm) $\left[x_f\,y_f\,z_f\right]$ | −7.21 | −3.50 | −39.93 | −7.25 | −2.90 | −38.05 |
| **LC Tibia point** (mm) $[x_t\,y_t\,z_t]$ | 8.60 | −51.88 | 19.44 | 10.58 | −51.69 | 19.26 |
| **LC Femur point** (mm) $\left[x_f\,y_f\,z_f\right]$ | 1.09 | 0.22 | 30.18 | 1.75 | −1.67 | 30.09 |
| **MC Tibia point** (mm) $[x_t\,y_t\,z_t]$ | −17.92 | 53.39 | −20.65 | −15.95 | 53.53 | −20.37 |
| **MC Femur point** (mm) $\left[x_f\,y_f\,z_f\right]$ | −4.61 | −3.65 | −22.42 | −6.45 | −4.24 | −22.93 |
| **ACL length** (mm) | 32.85 | | | 33.05 | | |
| **PCL length** (mm) | 39.65 | | | 41.46 | | |
| **MCL length** (mm) | 102.33 | | | 100.79 | | |
| **LC length** (mm) | 80.72 | | | 78.77 | | |
| **MC length** (mm) | 30.19 | | | 31.11 | | |
| **Patello–Femoral Group** | | | | | | |
| $\mathbf{n}_f$ (rad) $\left[\theta_f\,\phi_f\right]$ | 0.17 | 0.31 | | 0.14 | −0.24 | |
| $\mathbf{q}_f$ (mm) $\left[x_f\,y_f\right]$ | 7.06 | 4.60 | | 5.95 | 9.56 | |
| $\mathbf{n}_p$ (rad) $[\theta_p\,\phi_p]$ | 0.10 | 1.18 | | 0.07 | 0.44 | |
| $\mathbf{q}_p$ (mm) $[x_p\,y_p]$ | −41.66 | 2.78 | | −45.02 | 5.55 | |
| **PL Tibia insertion** (mm) $[x_t\,y_t\,z_t]$ | 35.81 | −46.32 | −3.87 | 35.81 | −46.32 | −3.87 |
| **PL Patella insertion** (mm) $\left[x_p\,y_p\,z_p\right]$ | 7.80 | −13.04 | 0.61 | 7.80 | −13.04 | 0.61 |
| **PL length** (mm) | 80.48 | | | 80.48 | | |
| $\lambda_0$ (mm) | 2.64 | | | 1.48 | | |
| $\lambda_1$ (mm) | −0.07 | | | −0.78 | | |

As for the PF, a reference pose for the definition of the model parameters was defined midway on the flexion arc, i.e., at 55° of knee flexion. The origin and insertion areas of the PT were obtained in the same way as ligaments. The corresponding rigid link was defined as the fibre which connects the projection of the two centroids on the corresponding surface. The length $l_6$ of PT (Figure 2c) was the distance between the endpoints of this fibre on the tibia ($\mathbf{c}_{t6}$) and on the patella ($\mathbf{c}_{p6}$) at the PF reference pose (Figure 2b,c). The axis of the helical pair was obtained from the MRI-based surfaces as the line connecting the centres of two best fitting spheres approximating the anterior part of the femoral condylar surfaces. This axis was described by a point $\mathbf{q}$ and a unit vector $\mathbf{n}$ in both $S_f$ and $S_p$ (Figure 2b,d). The unit vector $\mathbf{n}$ was represented in $S_f$ and $S_p$ at the reference pose, obtaining $\mathbf{n}_f$

and $\mathbf{n}_p$. Points $\mathbf{q}_f$ and $\mathbf{q}_p$ were the intersection of the axis with the sagittal planes of $S_f$ and $S_p$ at the same PF reference pose and their distance was $\lambda_0$. The pitch $\lambda_1$ of the screw pair was obtained from the angle between the helical axis and the normal to the best fitting plane to the intercondylar space, obtained from the CT–based bone model. In this way, the axial translation of the patella is obtained as $\lambda_0 + \lambda_1\vartheta$, where $\vartheta$ is the difference between the instantaneous and the reference flexion angle of the patella (Figure 2d).

The spatial motion of the model can be obtained by prescribing the range of motion of the TF flexion component or specific flexion angles, while the other motion components are computed by solving the closure equations of the mechanism [14,35]:

$$\left\| \mathbf{c}_{fi} - \mathbf{R}_{ft}\mathbf{c}_{ti} - \mathbf{P}_{ft} \right\| = l_i \quad (i = 1 \ldots 5) \tag{1}$$

$$\mathbf{R}_{fp}\mathbf{n}_p = \mathbf{n}_f \tag{2}$$

$$\mathbf{R}_{fp}\mathbf{q}_p + \mathbf{P}_{fp} = (\lambda_0 + \lambda_1\vartheta)\,\mathbf{n}_f + \mathbf{q}_f \tag{3}$$

$$\left\| \mathbf{R}_{fp}\mathbf{c}_{p6} + \mathbf{P}_{fp} - \left( \mathbf{R}_{ft}\mathbf{c}_{t6} + \mathbf{P}_{ft} \right) \right\| = l_6 \tag{4}$$

where $\mathbf{R}_{ft}$ and $\mathbf{R}_{fp}$ are the matrices that describe the tibia–femur and patella–femur relative orientation, whereas $\mathbf{P}_{ft}$ and $\mathbf{P}_{fp}$ are the origins of $S_t$ and $S_p$ represented in $S_f$ respectively. Equations (1) and (4) impose a fixed length at the rigid links representing the ligaments, contacts and PT. Equation (2) makes the rotation axes on the femur and patella coincident. Equation (3) imposes the distance between $\mathbf{q}_f$ and $\mathbf{q}_p$. The model geometry, whose parameters are identified at first as previously described from MRI and CT-based bone models, is then adjusted through an optimization procedure based on genetic algorithms followed by a Newton method [14]. This optimization procedure iteratively refines the model geometry within a limited parameter domain, comparing the experimental motion to the motion of the model in order to reduce the squared differences between the two at the corresponding flexion angles. In this way the effect of data uncertainties on the model definition can be largely reduced: the model geometry that best fits the experimental knee joint motion and avoids mechanism singularities is determined within narrow boundaries for each model parameter variation, consistent with possible inaccuracies in the model parameter definitions. The endpoints of the ligament and contact links were in fact constrained to be closer than 2 mm to their first estimate. The inclination and distance of the helical pair axis were constrained to be less than 5° and 5 mm from the corresponding first estimate.

Validation is performed by comparing the optimized mechanism motion to the experimental motion measured through 3DV. Differences between the two are evaluated quantitatively by computing the mean absolute error (MAE) and the maximum absolute error (MAX) of each motion component. To understand the amount of errors ascribable to the 3DV, motion differences are also compared to its theoretical accuracy [36].

## 3. Results

The initial and refined model geometries are reported in Table 1. The motion obtained from the model (Figure 3) shows great adherence to the experimental motion for both the TF and the PF joints. Considering the rotations and displacements at the joints, the MAE for the TF is always smaller than 0.89° and 1.35 mm with a MAX of 4.54° and 3.09 mm (Table 2); as for the PF, the MAE is smaller than 2.56° and 0.95 mm and MAX smaller than 7.21° and 2.19 mm (Table 2). However, the motion obtained by the model (Figure 3) lie within the limits of the theoretical accuracy of the 3DV [36] in most of the cases. The PF joint flexion exhibits higher errors at the extremities of the range of motion and a higher MAE with respect to the other pose parameters.

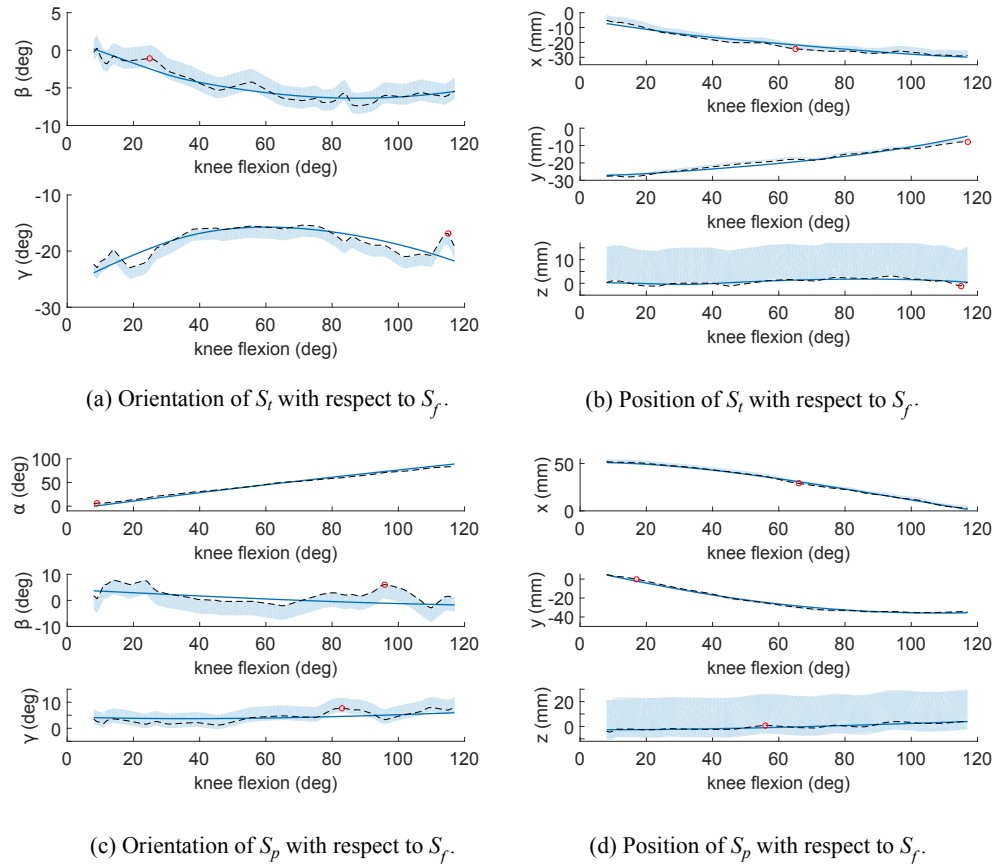

(a) Orientation of $S_t$ with respect to $S_f$.

(b) Position of $S_t$ with respect to $S_f$.

(c) Orientation of $S_p$ with respect to $S_f$.

(d) Position of $S_p$ with respect to $S_f$.

**Figure 3.** The subject-specific motion versus the TF flexion angle: experimental motion (black dashed lines) superimposed to the corresponding from the kinematic model (solid blue lines lines). Red circles highlight the point with the highest discrepancy between the experimental and the model motion. The pale blue areas around the experimental motion represent the theoretical accuracy of 3DV provided by Fregly [36]: they are centered with respect to the values of the experimental motion augmented by 3DV bias (i.e., the the mean value of the theoretical error between the 3DV and the experimental motion) with a width equal to the 3DV precision (i.e., the standard deviation of the theoretical error between the 3DV and the experimental motion).

**Table 2.** Mean absolute (MAE) and maximum absolute (MAX) errors of the pose parameters of the TF and PF joints.

| Group | | $\alpha$ (deg) | $\beta$ (deg) | $\gamma$ (deg) | $x$ (mm) | $y$ (mm) | $z$ (mm) |
|---|---|---|---|---|---|---|---|
| **TF joint** | MAE | | 0.47 | 0.89 | 1.34 | 1.05 | 0.54 |
| | MAX | | 1.44 | 4.54 | 2.63 | 3.09 | 1.75 |
| **PF joint** | MAE | 2.56 | 2.42 | 1.31 | 0.68 | 0.94 | 0.68 |
| | MAX | 5.80 | 7.20 | 3.09 | 1.86 | 2.19 | 1.92 |

## 4. Discussions

A procedure to define a subject-specific three-dimensional kinematic model of the knee joint starting from in vivo measures was presented and tested. A combination of standard medical imaging techniques such as CT, MRI and 3DV provided the data to customize an advanced model on a volunteer. An equivalent spatial mechanism was tailored on this subject and proved to replicate his knee joint motion, at the same time resulting consistent with the volunteer's joint constraints.

The acquired motion was a complete flexion-extension movement. Since the combined effect of hysteresis and fluoroscopy-related inaccuracies led to differences between the two phases,

the experimental reference motion was chosen as the mean between the two. While the in-plane motion components ($\alpha$ rotation, $x$ and $y$ displacements) were generally smooth, the out-of-plane components ($\beta$ and $\gamma$ rotations, z displacement) appeared noisy both in the original and in the mean motion (Figure 3), an issue that produced apparent cartilage surface interpenetration and separation, associated to unnatural ligament elongation. The same issues were absent in the natural motion obtained through the model: it was smooth and satisfied the joint constraints (both contacts and ligaments). This feature is of great importance, for instance in musculoskeletal models, where joint models are used to estimate the forces in the joint articular structures [37].

Since the model geometry obtained initially from the image data is refined by optimization to limit the effect of experimental inaccuracies and to overcome mechanism singularities, the definition of the parameter search domains is crucial. The isometric fibres were determined within the ligament bundles that have shown greater isometricity in previous investigations conducted with invasive experimental techniques [9]. Narrow boundaries were assumed here to keep consistency between the anatomical and the model constraints. These boundaries were chosen to be of the same order of magnitude of possible experimental inaccuracies during geometrical parameter definition. The boundaries for the helical axis of the PF group were slightly larger than the ones used for ligaments insertions, since the helical axis is less anatomically definite than the other model constraints. Optimization ended on or close to the boundaries in most cases. Larger boundaries could lead to a higher agreement between the model and the experimental motion, but this higher accuracy is paid at the expenses of the anatomical consistency of the model. This is particularly important when the model is used to evaluate the effect of articular constraints on the joint kinematics and dynamics. Moreover, an overfitting of the model on the experimental data has to be avoided, since, as discussed, these data could be affected by possible inaccuracies.

The overall good match between model and experimental motion (Figure 3) together with the close relation between model and anatomy show the reliability of the procedure and the accuracy of the model. In particular, errors (Table 2) are similar to those obtained with analogous models defined in-vitro with more invasive techniques [14], and are generally similar or lower than those obtained by other models that, however, are more computationally expensive [38,39]. Moreover, these errors in the present study can be accounted for to a number of reasons. The motion obtained by 3DV is generally affected by several inaccuracies [36,40]. 3DV in particular showed higher errors in the medio/lateral displacement, ab/adduction and internal/external rotations [40], though the technology of video-fluoroscopes has recently improved. This behavior seems confirmed also in the present study, by the differences between the acquired flexion and extension movements. These errors affect particularly the PF [36] due to the smooth and symmetrical shape of the patella. As shown in Figure 3, the difference between the experimental and the model motion lies within or in proximity of the boundaries of 3DV theoretical accuracy reported by Fregly et al. [36] in most cases, with the exception of patellar flexion. It is worth noting however, that these theoretical boundaries are ideal, since only the registration process was there considered, and thus they are actually narrower than those in a real scenario. This suggests that the proposed procedure identifies a kinematic model that acts as a sort of filter with respect to the fluoroscopy–related errors, accurately replicating the subject-specific motion and, at the same time, satisfying the anatomical constraints based on the subject anatomy.

As for the error in patellar flexion, which appears almost symmetrically placed with respect to the midpoint of the motion range (Figure 3c), this is related to the definition of the helical pair, which is less anatomically identifiable than the other model parameters and so needs further investigations.

This study has limitations. The procedure, here performed for the first time, was tested on a single non-pathological volunteer for ethical reasons. However, the study is intended to evaluate the feasibility of the procedure and, although it is not possible to assess features like the inter-subject variability, or the inter- and intra-operator variability, it is here shown that subject-specific spatial mechanisms representing both the knee joint kinematics and the anatomical constraints of a subject can be obtained in vivo. Application to a larger set of volunteers should not change the overall results

significantly: (a) the mechanism has already proved to be accurate in in vitro experiments [11,13,14]; (b) the imaging techniques are standard for many clinical procedures and should be applicable to any patient without particular changes; (c) the model solution and optimization were robust and no particular issues were found. Thus, the present procedure appears general and implementable in a variety of clinical scenario, and it could be further simplified by changing the measurement setup of the natural joint motion, used here as a reference for the model definition. 3DV was chosen to obtain the subject-specific motion: bi-plane 3DV could also be performed, but this would have implied a smaller overall motion of the lower limb, and higher doses to the volunteer [40]. As for the latter issue, in the specific case of single-plane 3DV methodology as that of the present study, the radiation dose exposure is generally more limited than using bi-plane systems; this might be further minimized in the future via more proper image acquisition angles, since it has been proved that the radiation doses may vary based on viewing angles of the fluoroscope [41]. The sensitivity of the model results with respect to the techniques used for MRI-CT and CT-3DV registration was not analysed. For instance, fiducial points were used in this study for MRI-CT registration, since they are standard, simple, and applied in many clinical procedures, in particular when aligning bony landmarks. However, many other techniques have been proposed, like mutual information and iterative-closest-point techniques [31,33]. The use of fiducial points for MRI-CT registration showed a good accuracy in a previous investigation [31], with a mean distance between corresponding surfaces lower than 2 mm, consistent with the accuracy of the medical imaging techniques used in the present study. Since one of the scopes of geometry optimization is to limit the influence of both the segmentation and registration inaccuracies, the effect of a different registration technique on the results is believed to be small. The joint model is rigid (i.e., no deformation is allowed to the ligaments) since it relies on the isometric behaviour observed for the ligaments during the joint natural motion [4,8–12]. However, rigid models are often considered in multibody simulations to reduce computational and numerical issues in dynamic analyses [2]. Moreover, as far as ligament elongation is of interest to discern the different joint behaviours under changing loading conditions, ligament and contact compliance can still be easily added in the model in a simple and computationally efficient way [42]. The joint natural motion was used as a reference for the model definition. It is indeed a simple and repeatable motion that can be easily performed within a fluoroscope. Moreover, it proved to be a good reference to represent also other physiologically loaded motions [43]. Finally, the proposed procedure required a measure of a reference motion for the model definition. However, it is firstly remarkable that the model is still able to describe the experimental motion with good accuracy despite the anatomical constraints are rigid and very close to the measured anatomy. Secondly, this is actually one of the features of the proposed procedure: it extends the standard method which imposes a given motion at the joint, by adding also information on the subject articular constraints in a simple and computationally efficient way.

In conclusion, the proposed procedure allows the definition in vivo of a subject-specific spatial model of the knee, including also the patello–femoral joint, which reproduces both the anatomical constraints and the natural joint motion of the subject. The accuracy and the overall validation were determined on a volunteer. The result is valuable since it allows the development of a methodological procedure to specialize the model on a specific patient, to be exploited in the future for personalized prosthesis design and medical procedures, as well as for the definition of more detailed, reliable and customized musculoskeletal models of the lower limb.

**Author Contributions:** Each author contribution to the article is subdivided as follows: conceptualization, N.S. and V.P.-C.; methodology, N.S., F.N., C.B. V.P.-C. and A.L.; software, F.N., N.S and M.C.; validation, F.N. and C.B.; formal analysis, N.S. and F.N. and M.C.; investigation, F.N., C.B., N.S., V.P.-C., A.L. and S.D.; resources, V.P.-C. and A.L.; writing–original draft preparation, F.N.; writing–review and editing, N.S., C.B., A.L. and M.C.; visualization, F.N.; supervision, V.P and A.L.; project administration, V.P.-C. and A.L.; funding acquisition, V.P.-C. and A.L. All authors have read and agreed to the published version of the manuscript.

**Funding:** This research received no external funding.

**Conflicts of Interest:** None of the authors have any conflict of interest with the subject matter of this manuscript.

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
