# Peer review of "An Anatomical-Based Subject-Specific Model of In-Vivo Knee Joint 3D Kinematics From Medical Imaging"

_applsci, doi:10.3390/app10062100_

Round 1

Reviewer 1 Report

The authors are presenting a method using static MR imaging and 3D-CT or imaging the knee and detect key points of a cinematic model already published in [13] and [14]. They validate the motion description obtained through these key-points by a 3D +T (or 3D-video) fluoroscopy.

The positions of the key-points and key-joins are specific to each patient, and their determination allow to derive a "personalized" model.

The CT images allow to determine the bony structure and the MR images allow to determine cartilage and attachment areas of the ligaments.

Some precisions should add value to the paper:

-regarding the image modalities: what is the Field intensity of the MRI (1,5 T or 3 T ? or ?)

-more details should be given about the 3D-video fluoroscopy

The key-points and key-joins are based on a manual registration of 3D CT surfaces to 3D MRI. The operator fix navigators on the 2 images and then a mean-square error minimization is achieved. That could be also discussed against mutual information maximization very meuch used in multimodal registration.

A central question related to this paper is to prove, or at least to give some clues, that the validation on only one patient could be safely generalized to any patient.

An important question that is not addressed in this paper, is how to minimize the dosis in the fluoroscopy acquisition. (see for example : D'Isidoro, Fabio, et al. "Determining 3D kinematics of the hip using video fluoroscopy: guidelines for balancing radiation dose and registration accuracy." The Journal of arthroplasty 32.10 (2017): 3213-3218. ).

Author Response

REVIEWER 1:

The authors are presenting a method using static MR imaging and 3D-CT or imaging the knee and detect key points of a cinematic model already published in [13] and [14]. They validate the motion description obtained through these key-points by a 3D +T (or 3D-video) fluoroscopy.

The positions of the key-points and key-joins are specific to each patient, and their determination allow to derive a "personalized" model.

The CT images allow to determine the bony structure and the MR images allow to determine cartilage and attachment areas of the ligaments.

Authors:

Thanks for the positive overall feedback. We wanted to test in fact the ability of our current modeling knowledge in combination with available medical imaging to develop subject-specific knee models.

REVIEWER 1:

Some precisions should add value to the paper:

-regarding the image modalities: what is the Field intensity of the MRI (1,5 T or 3 T ? or ?)

-more details should be given about the 3D-video fluoroscopy

Authors:

Further details on these points have been added.

REVIEWER 1:

The key-points and key-joins are based on a manual registration of 3D CT surfaces to 3D MRI. The operator fix navigators on the 2 images and then a mean-square error minimization is achieved. That could be also discussed against mutual information maximization very much used in multimodal registration.

Authors:

Important issues in fact. Two kinds of spatial registration were performed somehow. One registration was between CT and MRI data, to merge the geometry of the bones from the former and both cartilage and ligaments from the latter; the scope was to define the 3D anatomy as input for the kinematic model. The other registration was between the 3D surface model of the bones and the 2D images from videofluoroscopy, image by image; the scope was to reconstruct 3D kinematics of the joint for validation of joint motion from the knee model. We agree that many techniques were proposed for both registrations. The techniques used in the paper are standard, simple, and applied in many clinical procedures: these characteristics drove our choice. Moreover, the scope of parameter optimization performed in the proposed procedure is to reduce potential errors related to both the segmentation and registration, so we believe that the effect of different registration techniques on the results should be mitigated. All this is now more clearly represented.

REVIEWER 1:

A central question related to this paper is to prove, or at least to give some clues, that the validation on only one patient could be safely generalized to any patient.

Authors:

Of course more subjects will be necessary to strengthen the conclusions, but from the modeling and imaging points of view the present exercise is sufficient to say that the overall patient-specific modeling is feasible and can provide very satisfactory results. Nevertheless, this is now discussed better in Discussion.

REVIEWER 1:

An important question that is not addressed in this paper, is how to minimize the dosis in the fluoroscopy acquisition. (see for example : D'Isidoro, Fabio, et al. "Determining 3D kinematics of the hip using video fluoroscopy: guidelines for balancing radiation dose and registration accuracy." The Journal of arthroplasty 32.10 (2017): 3213-3218. ).

Authors:

Thanks, this is a relevant issue in fact. We were aimed at assessing the feasibility of the process, and we have done this in the best of the currently available imaging technologies and possibilities. Having obtained encouraging results, we can now test other less robust techniques, with less invasive scans but also, likely, with lower quality of the images and thus of the anatomical and mathematical models.

Reviewer 2 Report

The paper is well written and presents a procedure for defining a customized kneed model.

  1. Table 1 presents the values of the parameters used to define the knee model. Please introduce the set of mathematical equations of the knee model.
  2. Please present mode details on how you compute the first estimation and the optimized model.
  3. The title of Section 2 is “Methods”. It is unclear what exactly is presented in this section. What type of methods are presented? Which methods are novel?
  4. Please present arguments for why the model is validated for a single case. Why not search for more volunteers?
  5. Page 8, line 227. “In particular, errors (Table 2) are similar to those obtained with similar models defined in–vitro with more invasive techniques [14].” Please introduce a comparison of your model with other state-of-the-art methods. Why is the proposed model between then other models?
  6. Please clearly motion in the manuscript what is the proposed novelty of this work and how it distinguishes from other works.

Author Response

REVIEWER 2:

The paper is well written and presents a procedure for defining a customized kneed model.

  1. Table 1 presents the values of the parameters used to define the knee model. Please introduce the set of mathematical equations of the knee model.

Authors:

Ok, very reasonable request; these are now provided.

REVIEWER 2:

  1. Please present mode details on how you compute the first estimation and the optimized model.

Authors: Also these details are now provided.

REVIEWER 2:

  1. The title of Section 2 is “Methods”. It is unclear what exactly is presented in this section. What type of methods are presented? Which methods are novel?

Authors:

The Section “Methods” provides as usual the material used and the techniques adopted to achieve the current results. These are about data acquisition, medical image analysis, knee joint modeling, etc. The Results then report what has been obtained. These techniques not necessarily are all novel or original; what must be original is the overall exercise, which is the development of a successful subject-specific knee model starting from standard medical imaging. These aspects, and in particular the level of originality of these techniques are now emphasized more, in the Introduction, Methods and Discussion sections.

REVIEWER 2:

  1. Please present arguments for why the model is validated for a single case. Why not search for more volunteers?

Authors:

Of course the more are the subjects to whom the overall technique is applied, the stronger and more convincing would be the overall technique and the conclusions reported here. But, first of all, a successful application to a single subject can already provide demonstration of good potentials in feasibility and accuracy. Several parts of this technique had to be refined, and this is presented in the present work, also for other researchers to make progresses in this direction. It is also necessary to be clarified that this procedure implies irradiating the subject, and this rises relevant issues to us and to the Ethical Committee, beyond the availability of volunteers. This is now better discussed in Discussion.

REVIEWER 2:

  1. Page 8, line 227. “In particular, errors (Table 2) are similar to those obtained with similar models defined in–vitro with more invasive techniques [14].” Please introduce a comparison of your model with other state-of-the-art methods. Why is the proposed model between then other models?

Authors:

Thank you, this point in fact shall have been described better. The present exercise develops this knee model with the least invasive techniques available, because of the living subject involved. In cadaver specimens, i.e. in-vitro, as frequently used for similar exercises, bone pins can be implanted, more imaging can be collected etc. The concept expressed here was that very encouraging results have been obtained here, despite the fact that these restrictions to data collection were applied. Moreover, comparison with other state-of-the-art methods show that, despite its simplicity, our model replicates the volunteer kinematics with a generally higher accuracy. These points are now discussed better in Discussion.

REVIEWER 2:

  1. Please clearly motion in the manuscript what is the proposed novelty of this work and how it distinguishes from other works.

Authors:

Thank you, this is a sensible request, as we may have given for granted a few fundamental original steps forward in the present experimental and modeling study. These are now discussed more carefully in Discussion.

Round 2

Reviewer 2 Report

The authors presented a solid argumentation for my comments, and they modified the manuscript as I requested.